# Correction of Heritable Epigenetic Defects Using Editing Tools

**DOI:** 10.3390/ijms22083966

**Published:** 2021-04-12

**Authors:** Tayma Handal, Rachel Eiges

**Affiliations:** 1Stem Cell Research Laboratory, Medical Genetics Institute Shaare Zedek Medical Center, Jerusalem 91031, Israel; taymaha@szmc.org.il; 2School of Medicine, The Hebrew University, Campus Ein Kerem, Jerusalem 91120, Israel

**Keywords:** secondary epimutations, repeat associated diseases, genetic editing, epigenetic editing, transcriptional editing, DNA methylation, histone modifications

## Abstract

Epimutations refer to mistakes in the setting or maintenance of epigenetic marks in the chromatin. They lead to mis-expression of genes and are often secondary to germline transmitted mutations. As such, they are the cause for a considerable number of genetically inherited conditions in humans. The correction of these types of epigenetic defects constitutes a good paradigm to probe the fundamental mechanisms underlying the development of these diseases, and the molecular basis for the establishment, maintenance and regulation of epigenetic modifications in general. Here, we review the data to date, which is limited to repetitive elements, that relates to the applications of key editing tools for addressing the epigenetic aspects of various epigenetically regulated diseases. For each approach we summarize the efforts conducted to date, highlight their contribution to a better understanding of the molecular basis of epigenetic mechanisms, describe the limitations of each approach and suggest perspectives for further exploration in this field.

## 1. Introduction

Epigenetic modifications are heritable changes to the chromatin that do not include alterations in the DNA sequence. By regulating the physical structure and accessibility of the DNA, these modifications can switch genes off and on by dictating chromatin conformation either to a transcriptionally active (euchromatin) or silent (heterochromatin) state. Epigenetic marks mainly include DNA methylation and post transcriptional histone-tail modifications. These are obtained through the activity of a range of enzymes and chromatin interacting factors that read, write or erase specific DNA and histone modifications (for a comprehensive review on epigenetic modifiers, see reference [1].

DNA methylation, which is generally associated with gene silencing and repression, is an epigenetic mark that covalently occupies cytosine bases with a methyl group in mammalian cells at CpG sites. It is involved in repressing gene transcription in developmentally regulated and tissue-specific genes [2,3,4], suppressing transcription from repeat elements and transposons [5], regulating the mono-allelic expression of imprinted genes [6] and controlling X-chromosome inactivation in females [5]. This modification is chemically and biologically stable and is carried out through the counteracting activities of methylating (DNA methyltransferases, DNMTs) and demethylating enzymes (the Ten-Eleven Translocation enzymes, TETs) [7,8,9].

Apart from DNA, histone tails can also accumulate epigenetic marks; these mostly include the acetylation of lysine residues and methylation on arginine or lysine residues on histone H3 and H4 proteins. These types of modifications are determined by the coordinated activity of histone writers (histone methyltransferases (HMTs) and histone acetyltransferases (HATs)), histone erasers (histone demethylases and histone deacetylases (HDACs)) and histone readers (like HP1 and MECP2) [10]. Whereas acetylated histones always cause the chromatin to be competent for activation, methylation on histone tails can either promote (H3K4, H3K36 and H3K79) or repress (H3K9, H3K27 and H4K20) transcriptional activity, depending on its position [11].

It is important to note that DNA methylation and post -translational histone modifications appear to work in tandem. Specifically, open or active chromatin is associated with unmethylated DNA and active histone marks whereas closed chromatin is associated with methylated DNA and repressive histone- tail modifications. Together, they constitute the epigenetic memory, which is heritable from mother to daughter cells and in some loci is preserved across generations. However, one of the major differences between these two types of modifications is that DNA methylation promotes stable, long-term repression, while histone post- translational modifications are effortlessly reversible [12]. Because epigenetic modifications play an important role in the regulation of many genes, including those that are developmentally regulated, defects in their setting or maintenance commonly result in the incorrect expression of silenced genes or vice versa. Therefore, these types of abnormalities, which are also termed epimutations, are the basis of a long list of heritable diseases (see review by Zohgbi H et al., 2016 [13]).

Epimutations can be classified on the basis of their origin into primary and secondary epimutations. Primary epimutations are abnormal epigenetic changes that occur without any change in the DNA sequence. One good example is imprinting center defects in rare individuals with PWS [14]. By contrast, secondary epimutations are abnormal epigenetic changes that occur as a result of a change in the DNA sequence. This may stem from a somatic mutation, as in specific types of cancers, or from a germline transmitted or a de novo mutation, which leads to various epigenetically regulated developmental conditions. In the case of secondary epimutations, the underlying mutation can take place in a cis-regulating element or a trans-acting factor. While cis-acting mutations have a local effect by changing the epigenetic status and transcription activity of a specific locus, mutations in trans-acting chromatin modifiers have a more global effect by disrupting the transcriptional activity of many genes spread throughout the genome.

Because secondary epimutations are transmitted through cell generations, they comprise a main target for editing. With the development of editing tools, it has become possible to reverse or overcome epigenetic changes triggered by disease causing mutations. This can be accomplished either through the correction of the underlying mutation, or by rewriting/overcoming the epigenetic marks that are wrongly elicited in the genome. Together with use of programmable DNA binding and nicking platforms, particularly with the CRISPR-Cas9 system, this field of research constitutes a powerful paradigm for deciphering the epigenetic and developmental mechanisms of epigenetic diseases. This short review summarizes and discusses the efforts to date to correct secondary epimutations using a range of editing tools. Since nearly all attempts in human cells to repair secondary epimutations relate to repeat associated pathologies, this review concentrates on the correction of these types of mistakes in repeat associated loci. The application of these strategies for therapy or to other disease associated loci is beyond the scope of this manuscript.

### 1.1. Correction of Epimutations by Gene Editing

To date, only a handful of reports have documented attempts to reverse epimutations by correcting the underlying mutation. Oddly, all experiments have focused on the correction of epimutations that reside in, or act on, repetitive elements. The first study dealt with the deletion of a pathogenic GAA repeat expansion from the frataxin (*FXN*) gene in cells from patients with Friedreich ataxia (FRDA, OMIM#229300)). Friedreich ataxia (FRDA) is an autosomal-recessive neurodegenerative movement disorder that is caused by insufficient FRATAXIN protein. Most FRDA patients suffer from reduced levels of FRATAXIN due to a GAA tri-nucleotide microsatellite repeat expansion in intron 1 of the *FXN* gene. When the GAAs expand and reach the pathogenic range (>90 repeats), they incorrectly elicit heterochromatin in the region that surrounds the repeats because of the gain of repressive epigenetic modifications; DNA methylation and histone H3K9me2. This results in an *FXN* insufficiency due to reduced transcription initiation [15] and elongation [16]. The precise mechanism by which the GAA expansion triggers repressive epigenetic marks in the locus is unknown, although CTCF is most likely involved [17].

In an early work by Li and colleagues, GAA repeat expansions ranging from 630 to 1400 repeats were deleted from the *FXN* gene in patients’ lymphocytes and fibroblasts [18]. Using zinc finger nuclease-mediated gene editing, they induced a pair of double strand breaks (DSBs) that led to the induction of a large deletion which covered the entire repetitive region along with 1.2 kb flanking regions from intron 1. Although only one allele was successfully targeted, it led to the gain of active histone modifications (H3K9ac and H3K14ac) and to a 2.5–4.5 fold increase in *FXN* mRNA levels. The correction of the mutation not only increased *FXN* mRNA levels and upregulated protein expression with the change in histone modifications, but also improved the molecular phenotype of the disease. One potential caveat to this study was that the excision of the GAA repeats was accompanied by a large deletion of unrelated intronic sequences that flank the repeats, raising a concern as to whether the activity of other cis elements that reside in the deleted region was abolished by gene manipulation. Nonetheless, this early study suggested that it might be feasible to rescue the phenotype of a wide variety of epigenetically regulated conditions in humans by eliminating the causative mutation in the DNA sequence.

In a different study CGG repeats were successfully removed from the *FMR1* gene in iPSCs with the fragile X syndrome (FXS) mutation [19]. Fragile X syndrome (FXS, OMIM#300624) is the most common heritable form of cognitive impairment, and is the leading known genetic cause of autism. This X-linked inherited syndrome results from a deficiency in the fragile X mental retardation protein (FMRP) due to a tri-nucleotide CGG repeat expansion in the 5′-UTR of the X-linked *FMR1* gene [20]. When the CGGs expand and reach the pathogenic range (>200 repeats) they induce hypermethylation at the 5′-end of *FMR1*, including the promoter sequence [21]. This is coupled with a change from active (H3K4me3) to repressive (H3K9me2/3, H3K27me3) histone modifications, resulting in epigenetic transcriptional silencing of the gene, the cause of the FMRP deficiency in patients’ cells. Using XY FXS iPSCs which harbor more than 450 CGG copies and a completely hypermethylated and transcriptionally inactive allele, Park et al. [19] applied the CRISPR/Cas9 system to remove the CGG expansion from the gene. By creating a DSB upstream to the CGGs with a single gRNA, they induced somewhat random, but not overly large deletions (nearly 100 bp of flanking sequence) that spanned across the repeats. The successful elimination of the expansion in two out of three iPSCs clones led to a near complete ablation of abnormal methylation at the promoter (22 CpG sites) and changed the chromatin structure in that region by replacing repressive (H3K9me3) with active (H3acetylation and H3K4me) histone modifications. This resulted in the re-activation of the *FMR1* gene, reaching mRNA levels that are comparable with those seen in the wild type control. Furthermore, by directing the edited FXS iPSCs to differentiate into mature neurons, persistent expression of *FMR1* mRNA and protein levels were achieved even after prolonged differentiation. Overall, this study provided a proof-of-principle that epimutations can be reversed by the repair of the underlying disease-causing mutation. In addition, it implied that the mutation is not only critical for initiating heterochromatin but is also vital for its maintenance. One disadvantage to this study is that the repair was inaccurate and could potentially induce meaningful deletions from the upstream flanking region that may abolish the activity of other regulatory sequences.

In a parallel study by Xie et al., the CGG repeat expansion was removed from both, somatic cell hybrid CHO cells containing the human fragile X chromosome and from human XY FXS iPSCs in a more controlled fashion [22]. The expansion was deleted by targeting the Cas9 with a pair of gRNA to the immediate flanking regions relative to the repeats. Here again, the precise deletion of the FXS disease-causing mutation resulted in the re-activation of *FMR1* transcription and demethylation in five out of the nine gene edited hybrid clones, while in FXS iPSCs only one out of the five FXS iPSC gene re-activated clones transcription was successfully restored (8 CpG sites). The remaining iPSCs with a CRISPR cut deletion had relatively high methylation levels in the upstream flanking region, implying that epigenetic resetting is not efficient at all times. One explanation for this failure may be the quality of the iPSCs; the cells may have not been fully reprogrammed and retained epigenetic memory which might be more difficult to erase [23]. Alternatively, there may have been abnormally high levels of DNMT1 activity in these unusual repeat-less iPSCs, which could efficiently preserve already established methylation patterns. It would be important to repeat these experiments in slow-dividing cells such as neurons, to examine whether the rate of cell cycle progression affects the efficiency of epigenetically remodeling the locus.

In works that involved extending these studies to other diseased loci, Pribadi and colleagues [24] attempted to remove methylation in the *C9orf72* locus by gene editing in C9-related Amyotrophic lateral sclerosis and/or frontotemporal degeneration (ALS/FTD). Amyotrophic lateral sclerosis (ALS, OMIM #105400) is characterized by progressive muscle weakness and atrophy due to the degeneration of upper and lower motor neurons in the brain and spinal cord. By contrast, frontotemporal degeneration (FTD, OMIM #600274) affects behavior and cognition, and is caused by the preferential loss of neurons in the frontal and temporal lobe cortices. The leading known cause of ALS-FTD is a GGGGCC repeat expansion in intron 1 of the *C9orf72* gene, between noncoding exons 1a and 1b (also termed the C9 mutation) [25,26]. Multiple mechanisms are most likely involved in C9/ALS-FTD, including RNA toxicity and the deposition of dipeptide inclusions by unconventional repeat associated non (RAN)-ATG translation (for review, see reference [27]). As in the case of FXS, large GGGGCC repeat expansions lead to abnormal hypermethylation of the repeats [28], which may spread to the upstream promoter region of *C9orf72* [27,29,30,31] and coincides with the abnormal gain of repressive histone marks (H3K9me3 and H3K27me3) [32,33]. Although the contribution of C9 hypermethylation requires more elucidation, there is some evidence that hypermethylation attenuates the accumulation of repeat-containing toxic RNAs by a reduction in transcriptional activity from the upstream promoter [30]. This alleviates intron 1 retention following neural differentiation. Accordingly, and distinct from FXS, abnormal epigenetic modifications are thought to play a neuro-protective role in C9-related ALS-FTD by restricting the gain-of-function mechanisms (RNA and RAN-translation products) that are elicited by the mutation [32,34].

By targeting the Cas9 to both sides of the GGGGCCs, expansions of 800–1050 repeat copies were precisely excised from heavily methylated *C9orf72* alleles in C9/ALS-FTD iPSCs [24]. Methylation analysis in all the successfully targeted clones provided evidence for the complete erasure of aberrant methylation levels. However, this was uncoupled from a change in overall *C9orf72* mRNA levels, as was expected. It would be crucial to examine the effect of demethylation on alternative promoter usage and determine whether it rescues intron 1 retention. This would further substantiate the potential role of *C9orf72* hypermethylation in mitigating the toxic effect of RNA/RAN-translation products in C9/ALS-FTD.

In a different work, Yanovsky-Dagan et al. monitored changes in the epigenetic status of the DM1 locus subsequent to the excision of a large CTG expansion in mutant hESCs and patients’ myoblasts with the myotonic dystrophy type 1 causing mutation [35]. Myotonic dystrophy type 1 (DM1, OMIM#160900) is an autosomal dominant muscular dystrophy that results from a trinucleotide CTG repeat expansion (50–>3000 triplets) in the 3′-UTR of the *DMPK* gene [36,37]. While DM1 is primarily mediated by RNA/protein gain-of-function mechanisms [38], it also features local DNA hypermethylation in its severest form (congenital DM1) [39]. Although the clinical significance of *DMPK* hypermethylation remains controversial, there is some evidence for an inverse correlation between aberrant methylation levels and transcription levels in the downstream neighboring gene *SIX5* [40] which has been implicated in several clinical aspects of the DM1 pathology [41,42]. To address the question of whether hypermethylation can be reversed by repeat excision in DM1, hESC lines with a 2000 CTG repeat expansion were gene- edited to compare methylation levels before and after repeat excision. Here again, the methylation levels declined sharply from 100% to practically 0% (26 CpG sites) and H3K9me3 enrichments were lost on the background of the mutant allele in the repeat-deficient DM1 hESC clones. This, together with the findings in FXS and C9/ALS-FTD in pluripotent stem cells, implies that each DNA replication cycle methylation patterns are newly established (de novo) rather than copied from the template DNA strand. Strikingly though, when the same experiments were replicated in patients’ myoblasts with a 2600CTG expansion, the methylation levels and H3K9me3 enrichments remained abnormally high. This occurred despite many population doublings in culture, and is inconsistent with the view that epigenetic remodeling is more effective in highly replicating cells.

While most studies have focused on the correction of a heritable mutation that alters the epigenetic landscape of a specific locus (cis elements), there is only one example of the correction of a germline mutation in a trans-acting epigenetic modifier (see Table 1). In that study, the researchers attempted to repair two different mutations that result in Immunodeficiency, Centromeric Instability, and Facial dysmorphism type 1 (ICF1, OMIM#242860) syndrome [43]. This rare autosomal recessive disease, which is characterized by facial dysmorphism, immunoglobulin deficiency and chromosomal instability, is caused by bi-allelic loss-of-function mutations in the gene encoding for the de novo methyl transferase enzyme DNMT3B. Patients with ICF1 exhibit DNA hypomethylation at numerous genomic regions, including pericentromeric satellite 2 and 3 repeats and subtelomeric regions, which account for accelerated telomere shortening and premature senescence. Interestingly, ectopic expression of the wild type gene in affected fibroblasts fails to rescue the hypomethylated phenotype of the cells [44]. One potential explanation is the timing of DNMT3B activity, which is generally restricted to the preimplantation stage and is responsible for the exclusive de novo methylation of the majority of the repetitive sequences in the genome [7,45]. With this in mind, Toubiana and colleagues monitored changes in DNA methylation patterns in ICF1 patient-derived iPSCs, which best represent the developmental stage at which DNMT3B enzymatic activity is at its utmost [43]. Focusing on repetitive elements, they first validated the hypomethylated status of pericentromeric, centromeric and subtelomeric repeats. Next, they monitored changes in the methylation status at those regions following the correction of the causative mutations in the iPSCs by homologous DNA repair (HDR) via CRISPR/Cas9 editing. Although only one allele was successfully targeted, gene repair rescued the normal methylation patterns at the pericentromeric (satellite 2), and centromeric repeats (NBL-1 and p1A12) soon after editing. This contrasted with the ineffective rescue experiments in patients’ fibroblasts, highlighting the importance of cell type/developmental timing in efficiently inducing epigenetic reprogramming. Nevertheless, the restoration of normal methylation patterns was less efficient in subtelomeric regions, was incompatible with the normal phenotype of heterozygous carriers, and was unable to rescue the accelerated telomere shortening and premature senescence phenotypes observed in the gene-corrected iPSCs following differentiation. It should be noted that inefficient methylation in the subtelomeric regions was associated with persistent epigenetic memory which yielded abnormally high levels of H3K4me3 marks. In addition, it was possible to show that the marked enrichments in H3K4me3 abolished DNMT3B recruitment to those specific, still hypomethylated regions. Altogether, the findings from this study imply that the rescue of epigenetic diseases with genome wide disruptions will demand further manipulation beyond mutation correction. 

### 1.2. Reversing Epimutations by Epigenetic Editing

An alternative approach to correcting epimutations that are secondary to disease-causing mutations is to directly target the epigenetic marks rather than repair the DNA sequence (Table 1). This can be achieved through epigenetic editing, which refers to the removal/deposition of a specific epigenetic mark in a given locus by recruiting the related modifying enzyme to the site of interest through the simultaneous expression of a specific gRNA and a fusion protein between the modifying enzyme and a catalytically inactive Cas9 (dCas9) [48,49,50,51]. For example, it is possible to induce/delete DNA methylation in the mammalian genome by recruiting a dCas9 fused to DNMT/TET [52,53,54]. Consistent with this, Liu et al. erased the hypermethylated status of the CGG repeats at the *FMR1* locus in FXS patient-derived iPSCs with a 500 repeat expansion [46]. To do so, they designed a single gRNA that targeted the many CGG repeats present in the mutant *FMR1* locus to recruit the demethylating enzyme TET1 by fusion with dCas9. Targeting the repeats efficiently demethylated the entire 5′-end of *FMR1*, increased H3K27ac and H3K4me3 and decreased H3K9me3 enrichment levels at the *FMR1* promoter. As a result, epigenetic *FMR1* silencing was abolished, restoring FMRP expression levels in FXS iPSCs and in in vitro differentiated neurons. This procedure had minimal off-target effects in that only 29 out of more than 1000 dCas9-TET1 bound sites presented significant demethylation. While *FMR1* expression was increased by 1500-fold, most of the off-target sites showed either none or up to 4-fold changes in mRNA levels. *FMR1* expression and demethylation were maintained for at least two weeks after inhibition of dCas9-Tet1 activity and rescued the characteristic electrophysiological abnormalities of FXS neurons. In addition, *FMR1* re-activation by epigenetic editing was successful on differentiated neurons, albeit to a lower extent in terms of targeting efficiency and the degree of re-activation. Nevertheless, more than 50% of the demethylated neural precursor cells injected into the mouse brain exhibited FMRP expression after three months, indicating that *FMR1* re-activation by demethylation can be preserved for a long period in vivo despite the continuous presence of the mutation. Clearly, this report represents a breakthrough towards realizing the therapeutic potential of epigenetic editing. However, there are several concerns that should be addressed to make this approach feasible. One major difficulty is that it requires the constitutive expression of the epigenetic editor (in this case TET1) in contrast to the direct approach, where the DNA sequence is irreversibly repaired by a hit-and-run targeting method through transient expression of the CRISPR/Cas9 system. Furthermore, unlike DNA methylation in *FMR1*, the removal of a single modification may not be sufficient to induce the desired chromatin changes that will reactivate/repress the pertinent genes. Other concerns relate to the off-target effects of the system, which depend on the recruitment of the epigenetic editor to the repetitive sequence in *FMR1* along with many unrelated CGG repeat elements distributed in the genome, resulting in up to a 4-fold increase in mRNA levels of several unrelated genes. In addition, whereas in FXS cells, multiple copies of CGGs considerably enhance the targeting efficiency, it may not be as effective in other loci which do not possess such long repetitive targetable elements. The alternative for this would be to recruit the epigenetic editor to multiple sites across the locus simultaneously. One major challenge would be to identify the critical elements beforehand that are essential for controlling chromatin structure in that region. Finally, targeting epigenetic editors to specific regions in the genome will not be beneficial if the epigenetic alterations are caused by mutations in trans-acting factors with genome- wide disruptions as in ICF1 (DNMT3B), FSHD2 (SMCHD1), SOTOS (NSD1), RETT (MecP2), and many other epigenetically regulated syndromes.

### 1.3. Circumventing the Effect of Epimutations by Transcriptional Editing

A different approach to epigenetic editing for restoring the normal activity of epigenetically regulated genes is by directly targeting transcription through the expression of dCas9 fused to a transcriptional activator/repressor domain (Table 1). For example, Haenfler et al. [47] fused a dCas9 to multiple VP16 transcriptional activator domains (dCas9-VP192) to drive the expression of *FMR1* in FXS hESCs with an 800 CGG expansion without altering the DNA sequence. In this case, targeting with a single gRNA directed against the repeats (as opposed to targeting the promoter region) robustly enhanced *FMR1* transcription levels in mutant hESCs with a transcriptionally inactive allele despite promoter and CGG methylation, indicating that mRNA transcription is not directly halted by DNA methylation or heterochromatinization. However, in this study, elevated mRNA levels were not sufficient to significantly increase FMRP expression, presumably due to the long CGG tract which impedes translation efficiency [55,56,57,58]. Taken together, although the epigenetic marks remain unaltered, this approach makes it possible to overcome the undesirable effects of abnormal modifications by relating to transcriptional activity rather than to chromatin structure. Once again, many of the difficulties that are posed by epigenetic editing apply equally to the selective re-activation/repression of transcription by editing, including the need to constitutively express the dCas9-fused complex in the cells, and target multiple sites at the promoter region at the same time, as well as the inability to cope with epimutations caused by mutations in trans-acting factors that act globally in the genome. One final concern that applies to both epigenetic- and transcriptional-editing in the context of noncoding repeat expansions pathologies which is circumvented by gene editing, relates to the augmentation of RNA/RAN-translation toxic gain-of-function mechanism(s) due to the increase in lengthy mRNAs levels [59,60,61].

## 2. Conclusions

Epimutations may result from heritable changes in the DNA sequence. These defects are the cause of a long list of epigenetically regulated pathologies that result from mis-expression of gene(s) due to inherited mutations that affect the chromatin structure. While in some pathologies the mutation only has a local effect by changing the activity of a cis-regulating element, in others the loss-of-function mutation in a trans-acting factor such as a chromatin modifying enzyme or a chromatin remodeling complex results in a global change in the epigenetic signature of the genome. Very little is known about how and when heritable mutations lead to epigenetic abnormalities. With the advent of recently developed editing tools, it should be possible to address some of these unresolved questions on the mechanism(s), timing and reversibility of epigenetic modifications that are secondary to disease causing mutations (Figure 1). Understanding those mechanism(s) holds great promise for tackling the epigenetic aspects of this class of diseases and for the development of new therapeutic approaches.

## Figures and Tables

**Figure 1 ijms-22-03966-f001:**
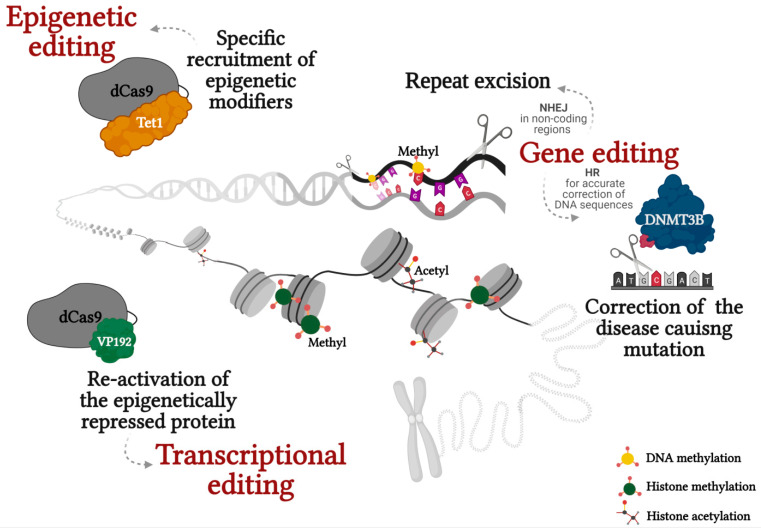
Correction of epimutations in repeat associated diseases through genetic, epigenetic and transcriptional editing. Created with BioRender.com.

**Table 1 ijms-22-03966-t001:** Current studies on correction of secondary epimutations in repeat associated loci.

Disease Name	Gene	Mutation	Epigenetic Modifications	Editing Method	Cells	Reference
Friedreich ataxia (FRDA)	Frataxin (*FXN*)	GAA expansion in intron 1	DNA methylation, H3K9me3, H3K27me3	Excision of the repeats by ZFN	FRDA lymphocytes and fibroblasts	[18]
Fragile X Syndrome (FXS)	Fragile X Mental Retardation 1 (*FMR1*)	CGG expansion in 5′-UTR	DNA methylation H3K9me2/3, H3K27me3	Excision of the repeats by CRISPR/Cas9 (NHEJ)	FXS IPSCs	[19]
				Excision of the repeats by CRISPR/Cas9 (NHEJ)	FXS IPSCs and somatic cells hybrids	[22]
				Recruitment of TET1 enzyme to repeats	FXS IPSCs and differentiated neurons	[46]
				Recruitment of VP192 to FMR1 promoter	FXS hESCs	[47]
Amyotrophic lateral sclerosis and/or frontotemporal degeneration (C9-ALS/FTD)	*C9orf72*	GGGGCC expansion in 5′-UTR	DNA methylation, H3K9me3, H3K27me3	Excision of the repeats by CRISPR/Cas9 (NHEJ)	C9/ALS-FTD IPSCs	[24]
Congenital Myotonic Dystrophy Type 1 (CDM1)	Dytrophia Myotonica 1 Protein Kinase (*DMPK*)	CTG expansion in 3′-UTR	DNA methylation, H3K9me3, H3K27me3	Excision of the repeats by CRISPR/Cas9 (NHEJ)	DM1 hESCs and Myoblasts	[35]
Immunodeficiency, centromeric instability, and facial dysmorphism type 1 (ICF1)	DNA Methyltransferase 3 Beta (*DNMT3B*)	bi-allelic missense mutations in the DNMT3B catalytic domain	DNA hypomethylation at pericentromeric, satellite 2 and 3 repeats, subtelomeric repetitive regions, H3K4me3	Correction of mutation by CRISPR/Cas9 (HR)	ICF1 IPSCs	[43]
				Ectopic expression of DNMT3B1 and DNMT3L	ICF1 fibroblasts	[44]

## Data Availability

Not applicable.

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
