# Peer review of "Correction of Heritable Epigenetic Defects Using Editing Tools"

_ijms, 2021, doi:10.3390/ijms22083966_

Round 1
Reviewer 1 Report
The article is very well -written. Just a few modifications are needed.
- please rephrase the title of the article. Sounds confusing.
- Flow of information in the abstract is quite abrupt please review the abstract accordingly to give a background, significance, rationale of your work, significance of your findings.
- please review line 32
- Format inconsistencies are present
Author Response
Based on the reviewer's comments provided to us, attached is the file that includes our response.

Reviewer 2 Report
Review report on “Repairing Epimutations With Editing Tools In Repeat Associated Loci” by Handal and Eiges. (ijms-1157735)
This is a review article on previous studies of correcting epimutations, one of the causes of human hereditary diseases, in cultured cells, such as iPS cells, by applying gene editing technology. The manuscript is easy to read, and the strengths and weaknesses of the approaches discussed are concisely described. Here are a few points that could help improve this paper.
-Abstract “by focusing on epimutations caused by, or relate to repetitive elements, primarily unstable noncoding repeat expansions”(Line 15)
In the main text, the authors state "Oddly, all experiments have focused on the correction of epimutations that reside in, or act on, repetitive elements" (line 93).
Therefore, it would be more accurate to clarify in the abstract that all previously reported studies were on epimutation related to repetitive elements.
-“This, together with the findings in FXS and C9/ALS-FTD in pluripotent stem cells, implies that each DNA replication cycle methylation pattern is newly established (de novo) rather than copied from the template DNA strand.”(Line 212)
In genomic imprinting, one of the best examples of long-term maintenance of DNA methylation, not all CpG methylations in the differentially methylated regions (DMRs) depending on parental origins are maintained, but the imprint control regions (ICRs) are persistently methylated, and peripheral methylations fluctuate. Therefore, I think it would be worth considering the possibility that excised CTG repeats in FXS may play a role similar to ICRs.
Author Response

(The authors gave the same response as above.)
